# Evolution Gym: A Large-Scale Benchmark for Evolving Soft Robots

**Jagdeep Singh Bhatia**
MIT CSAIL
jagdeep@mit.edu

**Holly Jackson**
MIT CSAIL
hjackson@mit.edu

**Yunsheng Tian**
MIT CSAIL
yunsheng@csail.mit.edu

**Jie Xu**
MIT CSAIL
jiex@csail.mit.edu

**Wojciech Matusik**
MIT CSAIL
wojciech@csail.mit.edu

## Abstract

Both the design and control of a robot play equally important roles in its task performance. However, while optimal control is well studied in the machine learning and robotics community, less attention is placed on finding the optimal robot design. This is mainly because co-optimizing design and control in robotics is characterized as a challenging problem, and more importantly, a comprehensive evaluation benchmark for co-optimization does not exist. In this paper, we propose Evolution Gym, the first large-scale benchmark for co-optimizing the design and control of soft robots. In our benchmark, each robot is composed of different types of voxels (e.g., soft, rigid, actuators), resulting in a modular and expressive robot design space. Our benchmark environments span a wide range of tasks, including locomotion on various types of terrains and manipulation. Furthermore, we develop several robot co-evolution algorithms by combining state-of-the-art design optimization methods and deep reinforcement learning techniques. Evaluating the algorithms on our benchmark platform, we observe robots exhibiting increasingly complex behaviors as evolution progresses, with the best evolved designs solving many of our proposed tasks. Additionally, even though robot designs are evolved autonomously from scratch without prior knowledge, they often grow to resemble existing natural creatures while outperforming hand-designed robots. Nevertheless, all tested algorithms fail to find robots that succeed in our hardest environments. This suggests that more advanced algorithms are required to explore the high-dimensional design space and evolve increasingly intelligent robots – an area of research in which we hope Evolution Gym will accelerate progress. Our website with code, environments, documentation, and tutorials is available at `http://evogym.csail.mit.edu`.

## 1 Introduction

One of the main goals of artificial intelligence is to develop effective approaches for the creation of embodied intelligent systems. Inspired from real organisms, where body structure and brain are two key factors for completing any task in a real environment, a successful intelligent robot typically requires concurrently optimizing its structure design and control mechanism. Such a co-design problem has been a long-standing key challenge in the robotics and machine learning communities. Surprisingly, despite its importance, most previous research works still either only develop complex control algorithms for existing robot structures [1, 2, 17, 30], or conduct co-optimization over robot morphology and control for only a few simple tasks (*e.g.*, running, jumping) [7, 14, 31, 32], especially

35th Conference on Neural Information Processing Systems (NeurIPS 2021).

in the soft body domain. The primary reasons behind the under-exploration of co-design algorithms in sophisticated problems are: (1) the underlying complex bilevel optimization scheme of a co-design algorithm, where the inner control optimization loop leads to a long iteration cycle of the whole optimization process; (2) the lack of a well-established benchmark platform providing the researchers with a suite to evaluate and compare different algorithms.

Digital benchmark environments have proven to be successful at promoting the development of advanced learning techniques via providing a comprehensive evaluation suite to make fair comparisons among different algorithms [5, 11, 36]. However, to our best knowledge, all existing benchmark platforms constrain their domains within control optimization problems, and the space of co-optimization environment suites is still rarely explored.

To fill this gap, in this work we propose Evolution Gym, a large-scale benchmark for evolving both the shape structure and controller of soft robots. The body of each robot in Evolution Gym is composed of various types of primitive building blocks (*e.g.*, soft voxels, rigid voxels, actuator voxels), and the control of the robot includes action signals applied on the actuator voxels. We choose to use this multi-material voxel-based structure as the representation of robot body since it provides a general and universal representation for various categories of robot designs, and at the same time results in a modular and expressive structure design space. We adopt a mass-spring dynamics system [26] with penalty-based frictional contact as the underlining physics engine. Such a light-weight simulator allows the co-design algorithms to significantly reduce the simulation cost and thus accelerate the develop-evaluate iteration cycle [3, 15, 23]. The back-end simulator is fully developed in C++ to provide further computing efficiency. Another feature of Evolution Gym is its large variety of tasks categorized by varying difficulty levels, which offer an extensive evaluation benchmark for comparing approaches. The benchmark is currently comprised of more than 30 tasks, spanning locomotion on various types of terrains and manipulation. Moreover, Evolution Gym is easy to use. In order to have user-friendly interfaces, we build a Python wrapper outside the C++ simulator and carefully design our APIs off of the well-received APIs of OpenAI Gym with minimum modifications. Evolution Gym will be released fully open-source under the MIT license.

In addition, we develop several baseline algorithms by integrating state-of-the-art design optimization approaches and reinforcement learning techniques. Specifically, in our baseline algorithms, design optimization methods are served in the outer loop to evolve the physical structures of robots and reinforcement learning algorithms are applied in the inner loop to optimize a controller for a given proposed structure design. We conduct extensive experiments to evaluate all baseline algorithms on Evolution Gym. The experiment results demonstrate that intelligent robot designs can be evolved fully autonomously while outperforming hand-designed robots in easier tasks, which reaffirms the necessity of jointly optimizing for both robot structure and control. However, none of the baseline algorithms are capable enough to successfully find robots that complete the task in our hardest environments. Such insufficiency of the existing algorithms suggests the demand for more advanced robot co-design techniques, and we believe our proposed Evolution Gym provides a comprehensive evaluation testbed for robot co-design and unlocks future research in this direction.

In summary, our work has the following key contributions: **(i)** We propose Evolution Gym, the first large-scale benchmark for soft robot co-design algorithms. **(ii)** We develop several co-design algorithms by combining state-of-the-art design optimization methods and deep reinforcement learning techniques for control optimization. **(iii)** The developed algorithms are evaluated and analyzed on our proposed benchmark suite, and the results validate the efficacy of robot co-design while pointing out the failure and limitations of existing algorithms.

## 2 Related work

**Robot co-design** Co-designing the structure (*i.e.*, body) and control (*i.e.*, brain) of robots is a long-standing key challenge in the robotics community. As the earliest work in this space, Sims [31] represents the structure of a rigid robot as a directed graph and proposes an evolutionary algorithm defined on graphs to optimize the robot design. Subsequently, the co-design of rigid robots is formulated as a graph search problem where more efficient search algorithms are applied [13, 27, 39, 41] to achieve increasingly interesting results. However, with the restriction of having rigid components only, these algorithms are unable to produce optimal or even feasible designs for many challenging tasks where a compliant joint or robot component is required to achieve the goal.

On the contrary, soft components offer much more flexibility to represent arbitrary shapes, making the design of more complex, agile, and high-performing robots possible. Inspired by this, some work has been conducted to co-design robots composed of soft cells. Cheney et al. [7, 8]; Van Diepen and Shea [37]; Corucci et al. [10] propose evolutionary algorithms to co-optimize the structure and control of voxel-based robots. However those algorithms typically parameterize the control as an open-loop periodic sequence of actuation, which prevents robots from learning complex non-periodic tasks such as walking on uneven or varying terrains. Spielberg et al. [32] and Medvet et al. [23] jointly optimize the spatial-varying material parameters and the neural network policy for soft robots but leave the shape of the robot fixed. Our proposed benchmark shares a similar expressive structure design space as Cheney et al. [7], but allows the control to be parameterized by a sophisticated neural network feedback policy. To handle such sophisticated joint optimization of the robot structure and high-dimensional neural network control policy, we develop several baseline co-design algorithms by combining state-of-the-art design optimization strategies and reinforcement learning techniques for control optmization.

**Benchmark environments for robotics learning** Present research in robotics learning is largely facilitated by emerging benchmark environments. For instance, OpenAI Gym [5], DeepMind Control Suite [36], rllab [11], and Gibson [40] have been developed to benchmark RL algorithms for controlling rigid robots. At the same time, PlasticineLab [16] is specifically designed for soft robot learning. However, the existing benchmark environments are all constructed for learning the control only. To enable the possibility of evolving the structure of a robot, the existing co-design work has to either implement their own testing environment [32, 7, 8, 10, 37], or make substantial changes on the underlying code of the existing control-only environments [29]. The independent development of testing beds requires non-trivial workload, and as a result, existing co-design works mainly focus on evaluating the robot on a few simple tasks such as walking on a flat terrain [7, 6, 8, 37, 32, 23], or swimming along a single direction [9, 39]. An unintended consequence of such independency is an indirect comparison among different algorithms. Evolution Gym fills this gap by presenting a large variety of tasks with different difficulty levels that span from locomotion to manipulation. The proposed benchmark suite can be effectively used to test the generalizability of the algorithms on different tasks, potentially accelerating research in robot co-design.

# 3 Evolution Gym

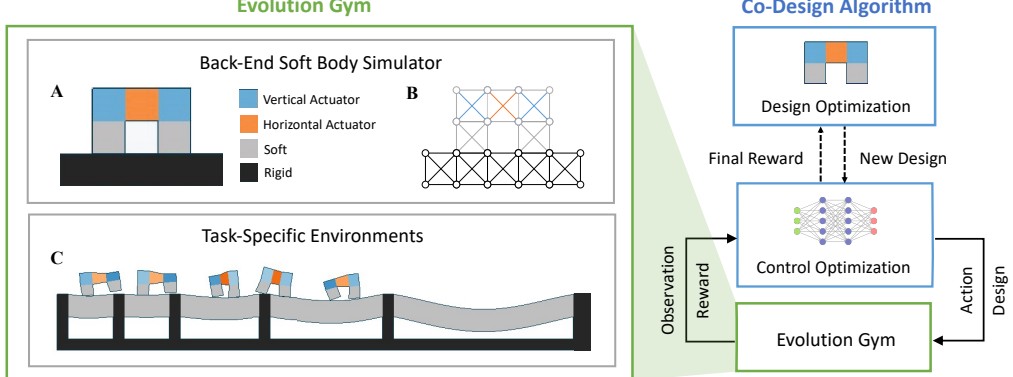

Figure 1: **Overview of Evolution Gym and its integration with the co-design algorithms.** Evolution Gym is comprised of a back-end soft body simulator (A, B) and task-specific environments (C). A user-customized co-design algorithm can be plugged in to optimize for both robot structure and control through interacting with Evolution Gym on a certain task.

## 3.1 Overview

In this section, we present Evolution Gym, a large-scale benchmark for the co-design of voxel-based soft robots. Evolution Gym is featured by its versatile and expressive multi-material voxel-based structure design space, flexibility of the controller parameterization, wide spectrum of tasks of various difficulty levels, fast back-end soft-body simulation support, and user-friendly Python interfaces.

As shown in the overview in Figure 1, Evolution Gym is comprised of a task-specific environment and a back-end soft-body simulator. The gym suite provides seamless interfaces with a user-defined co-design algorithm. The co-design algorithm typically consists of a design optimizer and a control optimizer. The design optimizer can propose a new robot structure to the control optimizer, then the control optimizer will compute an optimized controller for the given structure through interactions with Evolution Gym and finally return the maximum reward that this robot structure can achieve. In this way, Evolution Gym provides an easy-to-use platform for co-design algorithms to evolve both robot structure and control to optimize for robots' task performances. Evolution Gym is designed to be the first comprehensive testbed for benchmarking and comparing different co-design algorithms with the hope to facilitate the development of more novel and powerful algorithms in the co-design field.

## 3.2   Multi-material voxel-based representation

Evolution Gym employs a unified multi-material voxel-based representation for all the components in the environment (*e.g.*, robot, terrain, object) as shown in Figure 1A. Specifically, each robot in our gym is composed of rigid voxels, soft voxels, horizontal/vertical actuator voxels, and empty voxels. For terrain and objects, we use the same voxel-based structure but with passive voxel types (*i.e.*, soft/rigid voxels).

We chose a voxel-based representation for three main reasons. First, such a multi-material structure of robots provides a general and universal representation for various categories of robot designs and results in a modular structure design space. Additionally, with just the few voxel types described above, and less than 100 voxels per robot, we are able to construct a wide diversity of morphologies due to the resulting combinatorial robot design space. Even with this simple representation, our designed robots are capable of performing complex motions and completing difficult tasks. Finally, voxel-based robots can be simulated by a fast mass-spring simulation (see section 3.4) which allows our framework to be efficient enough to train robots in a matter of minutes and provides a computationally tractable benchmark for iterating co-design algorithms.

## 3.3   Task representation

Each task in Evolution Gym contains a robot structure proposed by the co-design algorithm, environment specifications (*e.g.*, terrain, object), and a task-related goal (*e.g.*, locomotion or manipulation). The tasks interface with the co-design algorithm through a few key elements including *robot structure specification*, *observation*, *action*, and *reward*. We introduce each element in detail below.

**Robot structure specification**   As described in Section 3.2, we construct each robot from primitive building blocks arranged on a grid layout. In code, each robot is specified as a material matrix of voxels $\mathcal{M}$ and a connection link list $\mathcal{C}$. The value of entry $m \in \mathcal{M}$ is a label corresponding to a voxel type from the set {Empty, Rigid, Soft, Horizontal Actuator, Vertical Actuator}. The connection link list $\mathcal{C}$ stores a list of connection pairs of adjacent voxels. The co-design algorithm can update the robot structure in the environment through *initialization* function with $\mathcal{M}$ and $\mathcal{C}$ as arguments.

**Observation**   The observation is composed in each step to inform the controller of state information of the robot, terrain information of the environment, and goal-relevant information. More specifically, let $N$ be the total number of voxel corner points of the robot. Then the state information of the robot in our tasks is a $(2N+3)$-D vector including the relative position of each voxel corner with respect to the center of mass of the robot ($2N$-D), and the velocity and orientation of center of mass (3-D). To handle complex tasks, specifically those with varying terrain types, an additional observation vector including terrain information is provided. We compile terrain information within a local window of size $2W$ around the robot into a length-$2W$ vector observation that describes the terrain's elevation. Furthermore, goal-related information is offered to inform the controller of the execution status of the current task. This goal-related observation is task-specific and is defined on each task separately. For instance, in manipulation tasks where the robot interacts with some object $O$, we provide orientation and velocity as well as the position of $O$'s center of mass relative to the robot.

**Action**   At each time step, an action vector from the robot's controller is provided to step Evolution Gym's simulator. In Evolution Gym, each component of the action vector is associated with an actuator voxel (either horizontal or vertical) of the robot, and instructs a deformation target of that

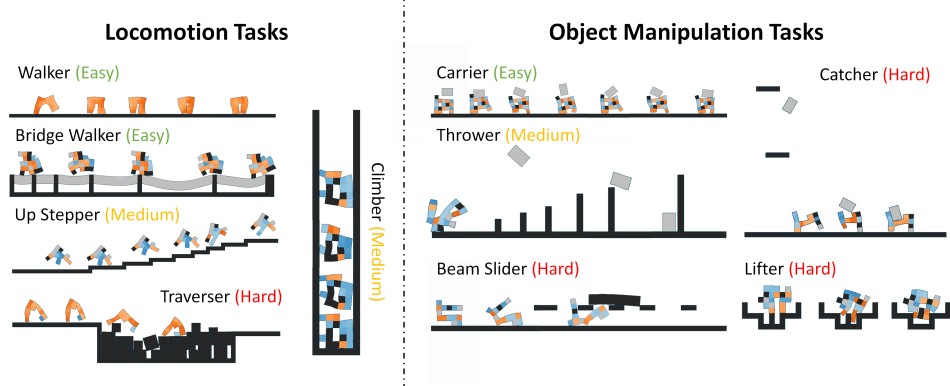

Figure 2: **A visual overview of selected 10 environments from Evolution Gym.** A verbal description of tasks is provided in Section 3.5.

voxel. Specifically, the action value $u$ is within the range $[0.6, 1.6]$, and corresponds to a gradual expansion/contraction of that actuator to $u$ times its rest length.

**Reward** Each task is equipped with a reward function measuring the performance of the current robot and the control action. The value of the reward is defined step-wise and is fed back to the agent through *step* function. The reward function is highly task-specific and should be defined to precisely characterize the robot's completeness of the task. Please refer to Section 3.5 and Appendix for detailed descriptions of the reward functions on each task.

### 3.4 Simulation engine

We model the dynamics of the underlying simulator as a 2D mass-spring system [26]. This simple, flexible formulation allows us to efficiently model soft robots with a wide range of capabilities in a wide range of environments. The simulation engine is written entirely in C++. We create Python bindings of our simulator so it seamlessly interfaces with standard learning frameworks.

The simulation represents objects and their environment as a mass-spring system in a grid-like layout (Figure 1B). Objects and their environments are initialized as a set of non-overlapping, connected voxels. On initialization, each voxel is a cross-braced square, but may undergo deformation as the simulation progresses. Each edge acts as an ideal spring obeying Hooke's law, with a spring constant defined by one of five possible material types. We employ symplectic RK-4 integration to step forward the simulation.

Collision detection is performed using a bounding-box tree structure [12]. Penalty-based contact forces and frictional forces are computed proportionally to the depth of penetration of the corresponding voxels in contact, and are applied on the voxel vertices in the normal and tangential directions of the contact respectively. Please refer to Appendix A for more details of simulation.

### 3.5 Benchmark environment suite

We have developed over 30 unique tasks with Evolution Gym and select 10 tasks here to illustrate the diversity and comprehensiveness of our benchmark task set. All tasks are organized into two categories – locomotion and manipulation – though some tasks are a mix of both. We further classify the tasks into different difficulty levels (*i.e.*, easy, medium, hard) based on the performance of the baseline algorithms (see Section 4) on them. We briefly introduce the selected tasks in this section. For more detailed descriptions and visualizations of the tasks, please refer to our website or Appendix B. It is also worth mentioning that our gym is designed to be extendable and the user can easily create new tasks for their needs.

#### 3.5.1 Locomotion tasks

**Walker** (Easy) This is a common standard task typically considered by previous works where the robot needs to walk on a flat terrain as fast as possible.

**Bridge Walker** (Easy) In this task, the robot traverses a series of soft "rope" bridges separated by fixed pillars, and similarly as before it needs to maximize its forward speed.

**Up Stepper** (Medium) The agent walks up a fixed staircase with steps of varying length.

**Climber** (Medium) The robot must climb two tall fixed walls on each side. The robot is rewarded by its upward climbing speed.

**Traverser** (Hard) In this hard task, the robot needs to traverse a pit of rigid blocks to get to the other side without sinking into the pit.

### 3.5.2 Object manipulation tasks

**Carrier** (Easy) The robot needs to catch a small, soft rectangular object initially dropped from above and then carry it along the forward direction. The robot is rewarded by the distance both it and the object have traveled.

**Thrower** (Medium) The robot throws a soft rectangular box as far as possible without moving itself significantly from its original position.

**Beam Slider** (Hard) In this task, a beam sits on top of a set of spaced-out floating platforms. The robot is rewarded for moving to the beam and sliding it in the forward direction.

**Catcher** (Hard) The agent needs to catch a spinning object randomly falling from a high location.

**Lifter** (Hard) The robot has to manipulate an object and lift it out of a hole.

## 4 Evolving soft robots

Robot evolution/co-design algorithms are formulated as a two-level optimization problem, which involves a design optimization method that evolves physical structures of the robots in the outer loop and a control optimization algorithm that computes an optimized controller for a given robot structure in the inner loop, as illustrated in Algorithm 1. We briefly introduce several instantiations of design optimization methods and control optimization methods in Section 4.1 and 4.2 that we use for evaluation on our benchmark, and more details can be found in Appendix C.

---

**Algorithm 1** Algorithmic framework of robot evolution

---

    **Inputs:** Task specification $T$, number of generations $n$, population size $p$.
    **Outputs:** The best robot design $D^*$ and controller $C^*$.
    $S \leftarrow \emptyset$                                  // Dataset of robot designs, controllers and reward
    $D_1, ..., D_p \leftarrow$ SAMPLEDESIGNS$(p)$          // Sample an initial population of robot designs
    **for** $i \leftarrow 1$ **to** $n$ **do**
       **for** $j \leftarrow 1$ **to** $p$ **do**
          $C_j \leftarrow$ OPTIMIZECONTROL$(T, D_j)$       // Optimize the controller of given robot design
          $r_j \leftarrow$ EVALUATEREWARD$(T, D_j, C_j)$ // Evaluate the reward of given design and controller
          $S \leftarrow S \cup \{(D_j, C_j, r_j)\}$             // Update the evaluation result to the dataset
       $D_1, ..., D_p \leftarrow$ OPTIMIZEDESIGNS$(S, p)$ // Optimize a population of robot designs to evaluate
    Find the best design $D^*$ and controller $C^*$ in dataset $S$ with the maximum reward $r^*$.

---

### 4.1 Design optimization

Design optimization aims at evolving robot structures to maximize the reward under two physical constraints: the body has to be connected, and actuators must exist. In this section, we introduce three instantiations of the design optimization algorithm (OPTIMIZEDESIGN in Algorithm 1).

**Genetic algorithm (GA)** GAs [24] are widely used in optimizing black-box functions by relying on biologically inspired operators such as mutation, crossover and selection, as demonstrated in previous works on evolving rigid robots [31, 39]. We implement a simple GA using elitism selection and a simple mutation strategy to evolve the population of robot designs. Specifically, in each generation, our elitism selection works by keeping the top $x\%$ of the robots from the current population as survivors and discarding the rest, where $x$ decreases gradually from 60 to 0 over generations. Next, we iteratively sample and mutate one of those survivors with $10\%$ probability of changing each voxel

of the robot to create more offsprings. Note that by mutating a voxel type from/to empty voxel, we are able to change the topology of the robot. The crossover operator is not implemented in our genetic algorithm.

**Bayesian optimization (BO)** BO [20, 25] is a commonly used global optimization method for black-box functions by learning and utilizing a surrogate model, which is usually employed to optimize expensive-to-evaluate functions, including evolving rigid robots in previous works [29, 21]. Specifically, we choose a batch BO algorithm as described in Kandasamy et al. [18] and implemented in the GPyOpt package [4] that supports categorical input data. We use Gaussian processes as the surrogate model, batch Thompson sampling for extracting the acquisition function, and L-BFGS algorithm to optimize the acquisition function. To ensure a fair comparison with other population-based evolutionary baseline algorithms, the batch size of this algorithm is set equal to the population size of other algorithms.

**CPPN-NEAT** CPPN-NEAT is the predominant method for evolving soft robot design in previous literature [6, 7, 8]. In this method, the robot design is parameterized by a Compositional Pattern Producing Network (CPPN) [33]. The input to a CPPN is the spatial coordinate of a robot voxel and the output is the type of that voxel. Therefore, by querying the CPPN at all the spatial locations of a robot, we can obtain the type for each voxel to construct a robot. At the same time the NeuroEvolution of Augmenting Topologies (NEAT) algorithm [34] is used to evolve the structure of CPPNs by working as a genetic algorithm with specific mutation, crossover, and selection operators defined on network structures. Our implementation of CPPN-NEAT is based on the PyTorch-NEAT library [28] and the neat-python library [22].

## 4.2 Control optimization

In this section, we introduce the specific control optimization algorithm (OPTIMIZECONTROL in Algorithm 1) that we use in the robot evolution algorithms. In previous works on evolving soft robots, the controller is either encoded as a fixed periodic sequence of actuation [7] or is parameterized as a CPPN that outputs the frequency and phase offset of the periodic actuation for each voxel [8]. However, the periodic pattern of the control prevents robots from learning complex non-periodic tasks such as walking on uneven or varying terrains. Therefore, we use reinforcement learning (RL) [35] to train the controller, making it possible for the soft robots to perform arbitrarily complex tasks in our benchmark. Specifically, we apply a state-of-the-art RL algorithm named Proximal Policy Optimization (PPO) [30] for control optimization of robots, with code implementation given by [19].

## 5 Experiments and results

In this section we present the evaluation results of baseline robot co-design algorithms on 10 selected benchmark tasks described in Section 3.5. The complete evaluation results on all our benchmark tasks can be found in Appendix E.

We develop three baseline algorithms for robot evolution by combing the three design optimization methods in Section 4.1 and PPO for control optimization in Section 4.2. Since the control optimization method is the same for all baseline algorithms, we simply use **GA**, **BO**, **CPPN-NEAT** to denote these three baseline algorithms with different design optimization methods. The evaluations of our baseline algorithms are performed on machines with Intel Xeon CPU @ 2.80GHz * 80 processors on Google Cloud Platform; GPU is not required. Evaluating one algorithm on a single task usually takes several hours to twenty hours, depending on the number of evaluations, size of population, etc. See Appendix D for more details on hyperparameters of all the experiments.

## 5.1 Comparisons among baseline algorithms

We plot the reward curves of the three baseline algorithms on 10 selected benchmark tasks in Figure 3. There is no single optimal algorithm that performs the best on all tasks, but overall, GA outperforms the other two baseline algorithms. This is surprising because our genetic algorithm is implemented with simple and intuitive operators for mutation and selection without sophisticated mechanisms. Therefore, we believe that with more carefully designed operators, GA has the potential to evolve much more intelligent robots. CPPN-NEAT generally performs well on locomotion tasks, as tested by previous works, but performs poorly on more complex manipulation tasks. This is possibly because

NEAT favors CPPNs with simpler structures, which encourages CPPNs to generate robots with more regular patterns. However, to succeed in complex manipulation tasks, some agile substructures of the robot must evolve, which might only exist in robots with irregular patterns. Finally, it is not surprising that BO performs poorly on most of the tasks because the high-dimensional categorical input parameter space and the noisy evaluation done by RL together pose a challenge to fitting an accurate surrogate model in BO.

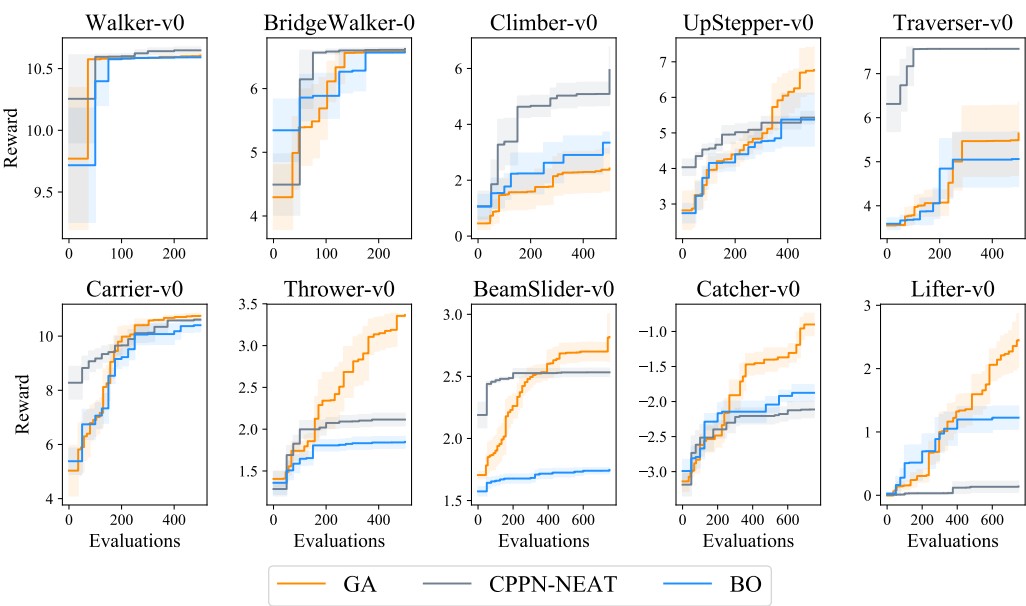

Figure 3: **Performance comparison among baseline algorithms.** We plot the best performance of robots that each algorithm has evolved w.r.t. the number of evaluations on each task. All the curves are averaged over 6 different random seeds, and the variance is shown as a shaded region.

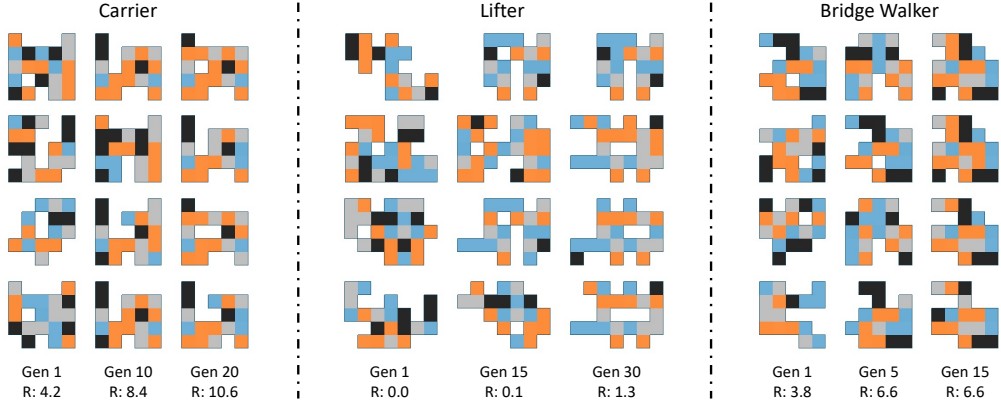

Figure 4: **Evolution of robot designs.** For each of the three selected tasks, we visualize the population in three different generations. Each column corresponds to one generation for which we show the four top performing robots along with their average reward.

## 5.2 Evolution analysis

In Figure 4 we visualize the top four robots in three different generations on training the genetic algorithm for the Carrier, Lifter, and Bridge Walker task. We also show the average reward these designs achieve.

In the carrier task, the robot must catch an object that falls from above and then carry that object as far as possible. Therefore, a successful design for this task achieves two main goals 1) allowing the

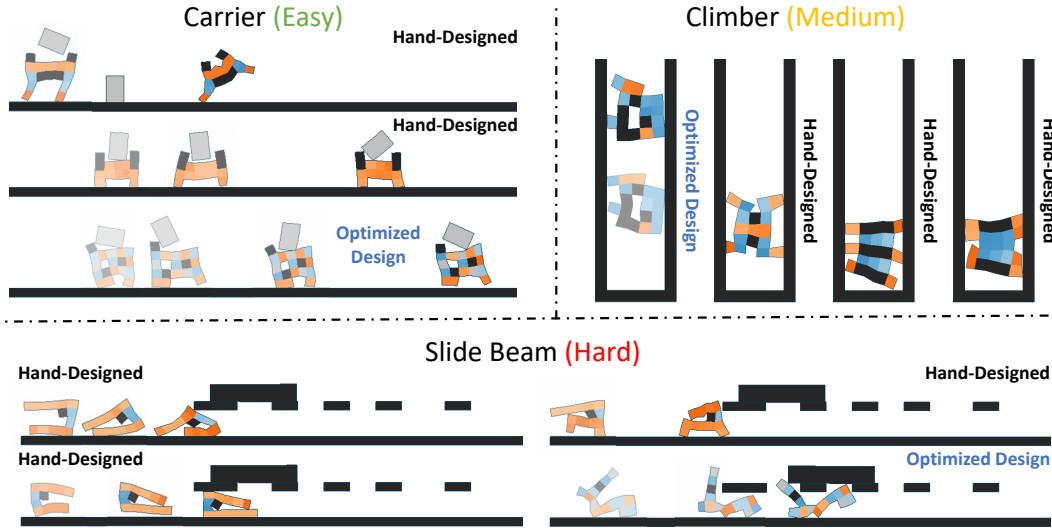

Figure 5: **Comparison between algorithm-optimized robots and hand designed robots on three tasks**. In each task, we visualize one robot optimized by the algorithm and several hand-designed robots.

robot to catch and hold the object securely 2) allowing the robot to move fast. We observe that robots with a block-holding mechanism and with legs are selected for in the top survivors of generation 1 (randomly initialized). As evolution progresses, these structures become increasingly optimized. Specifically, in later generations, the robots' structures allow them to walk faster while still preventing the block from falling.

A similar comparison pattern can be seen in the Lifter task, where the algorithm learns a parallel gripper-like shape underneath the robot in order to manipulate an object. Unlike in the carrier task, the design structures that the algorithm generates are not prominently found in the initial generation. Finally, these patterns are echoed in the Bridge Walker task. Here the robot learns to evolve a large front foot to maximize its surface area and friction force to best walk across the soft rope bridge.

## 5.3 Comparison against hand-designed robots

We compare the performances of robots optimized by algorithm and the hand designed robots on several tasks to show the necessity of a co-design algorithm (Figure 5). The structure of the hand designed robots are bio-inspired and manually constructed according to our best intuition, and their control are optimized by PPO.

For every task, the hand designed robots are outperformed by at least one algorithm (usually more). For instance, for the Climber task we tested numerous natural robot designs. However, none of them successfully climbed very far. The issue with our designs is that we could not find the right trade off between getting traction on the wall, and accelerating upwards. The genetic algorithm, however, is able to find this balance. It develops leg-like structures that help the robot make forward progress, as well as a long flat back that maximizes contact/frictional forces with the wall. Additionally, the genetic algorithm selects for having a hole in the center of its body, which helps it achieve a certain optimized walking motion.

For other tasks, the performance between the hand designed robots and the robots produced by the algorithms is quite comparable. This is the case with the Carrier robots, as a very natural hand-designed Carrier robot performs almost as well as the best optimized robots produced by the design-optimization algorithms.

In the final case, there are tasks where neither a hand designed nor robot produced by the algorithm could achieve satisfying performance. One such environment is the Beam Slider environment. For this task, many of the hand design robots fail to even achieve the first part of the goal and position themselves underneath the beam. While there is one robot produced by the genetic algorithm that does slide the beam across several pegs, from visual observation we believe it comes nowhere close

to exhibiting the optimal behavior in this environment. This suggests that further work is needed in designing co-optimization algorithms that can complete these hard tasks.

# 6 Conclusion and future work

In this paper we proposed Evolution Gym, the first large-scale benchmark for evolving the structure and control of soft robots. Through the wide spectrum of tasks in Evolution Gym, we systematically studied the performance of current state-of-the-art co-design algorithms. As a result, we observed how intelligent robots could be evolved autonomously from scratch yet still be capable of accomplishing some surprisingly complex tasks. We also discovered the limitations of existing techniques for evolving more intelligent embodied systems.

There are several potential directions to be explored in the future. First, with the help of our proposed benchmark, it is desirable to develop more advanced co-design algorithms to solve the difficult tasks which existing methods cannot address. Our currently implemented baseline algorithms share a bi-level optimization routine where the design optimization is in the outer loop while the control optimization is in the inner loop. However, Evolution Gym is agnostic to the specific training procedure used. As a result, some ideas for future work using our framework could include concurrently co-optimizing the design and control, neuroevolution algorithms, morphogenetic development, gradient-based methods for design optimization, or algorithms with decentralized controllers.

Second, a robot will be considered more successful if it can perform multiple tasks. Our benchmark suite naturally provides a comprehensive set of tasks and can potentially promote more exciting research work about multi-task or multi-objective robot co-design algorithms.

Another consideration is the specific morphological encodings used by the codesign algorithms as more intelligent encodings could lead to better performance. For instance, [38] analyzes the strengths and weaknesses of different morphological encodings. Our baseline algorithms use a direct encoding and CPPN but exploring other encoding representations remains interesting future work.

Finally, since tasks in Evolution Gym are currently limited to either locomotion or manipulation, we plan to further extend Evolution Gym to additional task categories such as flying or swimming by incorporating new simulation capabilities.

Overall, we believe our carefully-designed benchmarking tool fills an important missing piece in research in soft robotics and robotic evolution algorithms. Armed with the flexible and expressive framework Evolution Gym provides, we are optimistic that future researchers will use Evolution Gym as a standard test bed to improve co-design methods and evolve more intelligent robots.

## Societal Impact

We regard this work as a very preliminary piece of research in the field of soft robot co-design, and therefore think that we are still far away from causing harm to society. However, we can definitely foresee some problems if this technology were to be applied in the real world on a large scale. For instance, this work may inspire the automatic design of real biological creatures in which serious ethical issues exist. Additionally, since the users have full control over the reward design when customizing the benchmark environments, they could specify pernicious goals and encourage the co-design algorithm to produce more biased results.

## Acknowledgments and Disclosure of Funding

We thank Tao Du and the anonymous reviewers for their helpful comments in revising the paper. This work is supported by the Defense Advanced Research Projects Agency (FA8750-20-C-0075).

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
