# OpenReview forum: "Evolution Gym: A Large-Scale Benchmark for Evolving Soft Robots"
_NeurIPS.cc/2021/Conference — NeurIPS 2021 Poster_

### Official Review · Reviewer_VGKK · 2021-06-27

**Rating:** 7
**Confidence:** 3

**Summary:**

This paper introduces a simulator and suite of benchmarks for studying the co-design of soft-robots and their controllers. It provides a simplified abstraction to describe soft robots from their components, a corresponding physics engine, and a set of tasks and associated metrics to optimize.

**Ethical Concerns:**

None.

**Limitations And Societal Impact:**

See review.

**Main Review:**


This is a solid paper. It is well organized, describes the problem it aims to tackle clearly, and sufficient details about the various assumptions and simplifications that were made, the benchmark tasks, and a solid set of baselines to work from.

A few things I'd wished to see:

- more details about the system and computational requirements. As a user, I'd love to get a sense of what I'd be getting into if I were to attempt tackling such benchmarks.

- the code! The paper says the APIs are OpenAI-gym-like, and I would have loved to be in a position to evaluate the usability of the benchmark suite first-hand.

- Perception being considered as part of the design space. The agents have access to the full state of the robot and to a local window of the environment. This may be a case of 'too little and too much at the same time': many of the tasks mean the controller needs to reason about contact, which it only has access to indirectly through geometric reasoning from the state, which is hard. Enabling the robots to have access to summary state like contact information, or range sensors, could actually lead to improved designs, faster learning, and would also make the benchmark one step closer to being useful for more realistic embodiments which will never have access to the full state of the system, particularly if they're soft.

- Any suggestion that insights that are transferrable beyond this specific sets of benchmarks can be derived from this co-design problem setting. Can we learn controllers that facilitate the selection of candidate physical designs faster? Can we quickly validate candidate designs via simple controllers and iterate faster? Lots of possible questions to ask, and which would enhance the value of the overall benchmark suite.

None of those issues however detract from the fact that this is an interesting benchmark. I will flag my review as comparatively lower confidence because I don't know the landscape of the existing tooling around morphology design and soft robots, so I can not properly judge how unique this contribution is.

**Time Spent Reviewing:**

1

---

> ### Author Response · Authors · 2021-08-10
> **Authors' Reponse to Reviewer VGKK**
>
> Dear reviewer,
>
> Thank you for your feedback and constructive suggestions. We are pleased to hear you found our paper well-written and that you see the novelty of our methods. Below, we have addressed your comments in detail.
>
> **(1) More details about the system and computational requirements**
>
> As mentioned in our manuscript, we ran our experiments on machines with an Intel Xeon CPU @ 2.80GHz * 80 processors through the Google Cloud Platform. Note that a GPU was not required because our parallelization happens across processors. Evaluating one algorithm on a single task usually takes several hours to twenty hours, depending on the number of evaluations, size of the population, etc. We currently have bindings compiled for Ubuntu 18.04, Windows, and MacOS as well as a script that allows users to compile bindings for their particular system.
>
> **(2) Code release**
>
> Our code is currently available on our website at this [link](https://sites.google.com/view/evolution-gym-benchmark) and will be posted free and open-source on Github along with documentation upon publication of the paper.
>
> **(3) Perception**
>
> We agree that adding contact information and ranged sensors will be valuable additions to our work, and we will definitely include these as we continue to maintain Evolution Gym.
>
> **(4) Co-design strategies and other insights**
>
> Yes! We hope that our benchmark will allow future researchers to uncover these insights and answer exactly these questions.
>
> It was out of the scope of our paper to analyze all such strategies, but we hope future researchers will formulate similar ideas and evaluate them using our framework. We are optimistic that future researchers will use Evolution Gym to improve co-design methods, not just by tweaking existing algorithms slightly, but by fundamentally redesigning them so that they apply beyond our framework. We would be happy to include a discussion on this topic in our paper if you believe it would further our work.
>
> -------------------------------------------------------------------------------------------------------------
>
> Overall, our carefully designed benchmarking tool fills an important missing piece in research in soft robotics and robotic evolution algorithms. We believe NeurIPS would provide the perfect launching pad to expose our tool to the types of researchers equipped and excited to utilize such a benchmark.
>
> We hope in this response we have clarified the goals of our original paper and adequately addressed your concerns. Your comments have helped us strengthen the work. If you have any other questions, we would be open to further discussion.
>
> Sincerely,
>
> The authors of Evolution Gym

---

### Official Review · Reviewer_j4pt · 2021-07-01

**Rating:** 4
**Confidence:** 4

**Summary:**

The paper proposes a framework for benchmarking algorithms for optimising both bodies and controllers of soft-bodied robots. It also presents three optimisation solutions and evaluate them on a suit of benchmark tasks. The authors also evaluate some hand-coded solutions; the performance of these is worse for some tasks, while about the same for others.

**Limitations And Societal Impact:**

I think this is not applicable for this work.

**Main Review:**

The main novelty concerns the benchmarking framework, called Evolution Gym. The challenge here is to define representations that are sufficiently generic to define a range of environments and tasks, robot morphologies and controllers. This is addressed by extending existing approaching in meaningful ways.

The paper is overall well presented.

Major comments:

The Evolution Gym has merit in offering possibilities to benchmarking solutions. The robot morphologies and designs are fairly intriguing. Yet, I am not convinced Evolution Gym is a suitable tool for solving real-world problems with soft robotics. Evolution Gym is limited to 2-D. Even Sim's work from 1994 (the tasks of which are described as "simple" by the authors) was conducted in 3-D.  Another simplification is the need for discretisation (e.g. you cannot have a tilted plane as the ground). Moreover, the treatment of sensing seems rather simplistic. While "observations" are considered, it is not clear how these would be obtained in practice (e.g. position of corner modules relative to the barycentre of the soft-bodied robot). There does not appear to be noise on actuation or sensing. The only source of uncertainty - to my understanding - is the initial placement of the object in the manipulation tasks.

Real-world scenarios are also likely requiring a robot to deal with a range of object geometries to be manipulated. To my understanding the objects that are manipulated in the same task are of constant shape and size. On a similar note, it would have been interesting to define the framework more general, so that the same robot morphologies can be tested not on a single but a set of tasks.

Regarding Algorithm 1, why it is necessary to start the controller evolution from scratch every generation (as $S$ is only passed as an argument to OptimizeDesign, but not to OptimizeController). I understand that not every controller will be compatible with every morphology. However, there could be ways to deal with this.

This raises another question, why having an outer and inner loop at all for morphology and design optimisation? Why not co-evolve them as part of the same genotype? Maybe your approach offers better "evolvability"? In any case, it should be made clear that it is up to the user to define their optimisation solutions.

For the benchmarking framework to be widely adopted, it would be good to understand how it can be extended. Moreover, it would be good to have some tasks with scalable complexity, thereby allowing the community to progress along the same "axis", rather than each having to invent their own tasks.

The actions need to be explained more precisely. I guess that each component of the action vector is associated with an actuator voxel (not each "value" of the action vector as is stated).

I would have appreciated more detail on the controller design, and how it combines sensing with actuation (I only gave the supplementary material a quick check, "actions" are no mentioned therein). For example, there is a sensing window of size 2W providing terrain information; but up to which height, for example, for the Slider task in Figure 2, where the robot needs to perceive objects above it? Related to this, I wonder, which of the tasks, if any, require closed-loop control, and how is it realised? This I was missing from the descriptions in Section 3.3.

You say twice "general and universal representation for various categories", which is somewhat contradicting. How universal is the representation? You state that the shape of a robot can change as voxels can flip from/to empty; can the bounding box of the robot evolve as well? Can the building blocks be combined in arbitrary ways (e.g. horizontal actuator on top of vertical actuator) provided the structure remains connected? Why can objects not have empty cells (holes) - wouldn't this impact on inertial forces?

It seems that each optimisation is run only 3 times per task (Figure 3 states "3 different random seeds"). This would not be sufficient for a rigorous comparison.

I like the finding that a simple GA was possibly the best performing solution (though could be due to the limited sample size). This would show how important benchmarking is. However, there is the risk all the effort is spend on getting the perfect optimisation algorithm, whereas the real problems would relate more to how the robot can best sense its environment and move therein - and some of this is pre-determined/rigid given the representations that has been chosen.

The paper consistently talks about "optimal" control/designs. There is nothing to suggest that any of the solutions are optimal.

Minor comments:

It is stated that the "robots are evolved autonomously from scratch without prior knowledge", however, designing the reward function "should be defined to precisely characterize the robot's completeness of the task" at every step.

I would place Figure 1 on top to save place.

Typo - "Speilberg"

"combining the state-of-the-art design optimization strategies.." - I would remove "the"

"with of steps of"

I found the description of the algorithm first confusing, as the line with OptimizeControl is described as "Optimise the controller" whereas the one with OptimizeDesign is described ass "Propose [..] a robot design".

In Figure 3, the y axis ranges are not consistent, making it hard to compare the solutions.

UPDATED (following rebuttal):

I thank the reviewers for their responses. I am still concerned by the limited number of independent trials (3, or possibly 6 in the camera-ready version). This would seem to set low standards, especially for a benchmark paper. Moreover, it is not clear how increasingly complex problems could be designed. The authors state that "ranking [their] tasks by difficulty level achieves a similar purpose". However, with only a limited number of tasks, the complexity would be bounded, and the ranking is somewhat subjective.

I have changed the overall rating from 5 to 4.

**Time Spent Reviewing:**

5

---

> ### Author Response · Authors · 2021-08-10
> **Authors' Reponse to Reviewer j4pt**
>
> Dear reviewer,
>
> Thank you for your feedback and constructive suggestions. We are pleased to hear you found our paper well-written and that you see the novelty of our methods.
>
> First, we would like to clarify the intended purpose of our work. Our main contribution is to provide the first benchmark environment for evaluating and improving robot co-design algorithms. Our benchmark environment is designed to be efficient and effective through selecting a compact robot morphological representation (2D voxel-based soft robot) and implementing a fast simulation (C++-based mass-spring system). Such a lightweight benchmark helps reduce the time spent in the simulation during evaluating AI algorithms and accelerates the development iteration of the new algorithms. We hope that future researchers will use Evolution Gym as a tool to develop new, powerful co-design techniques that could not have been realized otherwise.
>
> Below, we have addressed your major comments in detail.
>
> **(1) 2D nature of Evolution Gym**
>
> Thanks for pointing out this. Our benchmark is designed to be a quick testbed for co-design algorithms. However, the co-design algorithms can be agnostic to whether the application is 2D or 3D (similarly as reinforcement learning algorithms for control optimization which are agnostic to whether the problem to be solved is 2D or 3D).
>
> Further, we believe the 2D nature of our environment is one of its strengths. 2D soft-body simulation provides enough complexity of the task while being much more computationally cheap compared to 3D soft-body counterparts. For instance, training a controller for a 3D soft robot on a simple task can take on the order of hours based on more realistic FEM simulation (see Du et al. 2021 [1]). By comparison, in our framework based on 2D mass-spring systems, the control policy of a single robot can be trained on a 4-core machine in a matter of minutes. With such speeds, we believe that future research endeavors in co-design and control optimization algorithms would be more computationally tractable.
>
> Finally, as evidenced by our experimental results, the state-of-the-art co-optimization algorithms fail to search successful robots on many of our hard tasks, even in a simple 2D environment. In other words, the limitations of the learning algorithms stopped us from increasing the sophistication of our environment. However, we believe that 3D extension is an important future work when the learning algorithms become more intelligent and sample efficient.
>
>
> **(2) Justifying the voxel-based design space and discretization in the simulation**
>
> We focus on voxel-based soft robots in this work with inspiration from real cell-based organisms in nature. Such discretized voxel-based design space provides us with the balance of expressiveness and efficiency needed for a benchmark suite. With only several different voxel types and < 100 voxels for each robot, we are able to construct a wide diversity of morphologies; many trained robots are able to perform complex motions and complete difficult tasks. Further, the voxel-based design enables fast simulation which allows our framework to be efficient enough to train these robots in a matter of minutes.
>
> **(3) Clarifying language around observations (sensing), actions, and the controller**
>
> Clarification on the action space: Each value of the action vector corresponds to an actuator voxel to control how the actuator voxel expands/contracts, as mentioned in Section 3.3. Specifically, if the action value > 1, then the corresponding actuator expands; if the action value < 1, it contracts.
>
> More details on the controller: First of all, our framework is agnostic to the type of controller used -- it’s compatible with either open-loop or closed-loop controllers. However, we chose to use closed-loop control (specifically, deep RL) in the experiments because it is a more general control strategy for both stochastic and deterministic tasks. For example, our closed-loop controller can handle the randomness in Catcher-v0 while the open-loop controller would be infeasible in this case. We use the standard implementation of PPO, a deep RL algorithm as the closed-loop controller in our experiments. For the sliding window, it will relay the distance to the nearest object above the robot, up to a max distance of 5 voxels. Thanks for pointing out these confusions and we will update the expression in the paper to make it clear.
>
> **(4) Sources of uncertainty (in sensing and object morphology)**
>
> Adding random variation in the geometry of manipulated objects is a great idea. We would be happy to include experiments with this change in a revised version of the paper. As we continue to maintain the Evolution Gym framework, we would strongly consider adding other forms of uncertainty, such as noise in sensing.
>
> However, for now, adding extra complexity to our treatment of sensing might detract from the main goal of the co-design algorithms and make them struggle unnecessarily. In fact, other successful benchmarks like Mujoco also use a simple sensing approach. This suggests that such an approach does not hinder algorithmic development. As co-design methods improve, we may be able to introduce more complicated sensing as well as real-world obstacles such as noise.
>
> **(5) Why do our baseline algorithms have separate loops for design and control**
>
> We implemented the state-of-the-art algorithms which are all in this separate evolution mode. Effectively re-using the controllers between designs is a great idea and would be an interesting work to explore in the future.
> We illustrate a separate-loop evolution framework in Algorithm 1 because our implemented baselines follow this routine. However, our benchmark suite is capable of more sophisticated co-design algorithms such as reusing controllers between designs.
>
> **(6) Tasks with scalable complexity**
>
> This is a good point, and it would certainly be possible to design tasks with scalable complexity. We believe that ranking our tasks by difficulty level achieves a similar purpose: allowing researchers to benchmark their algorithms without needing to invent tasks from scratch. Nevertheless, we would be happy to add this feature as we continue to maintain the framework.
>
> **(7) Number of experimental runs**
>
> Thanks for pointing this out and we totally agree with you. The only reasons that stopped us from testing more seeds before submission are the huge computation cost of co-optimization and the variety of our environments. Given more time, we will definitely update the paper and include the results from at least 6 different seeds.
>
> **(8) Use of the word “optimal”**
>
> We agree that our language is misleading. The goal of our research was not to evolve optimal designs but to show that even in simple 2D environments, current state-of-the-art co-design methods often fail to evolve successful designs. We will review our language and reduce or contextualize the use of the word “optimal” so as not to confuse readers -- for instance, we can replace “optimal designs” in many instances with “best evolved” or “best learned” designs.
>
> **(9) Minor Comments**
>
> We are happy to change the various grammar and formatting issues as suggested by the reviewer as well as clarify our language where appropriate.
>
> ----------------------------------------------------------------------------------------------------------
>
> Overall, our carefully designed benchmarking tool fills an important missing piece in research in soft robotics and robotic evolution algorithms. We believe NeurIPS would provide the perfect launching pad to expose our tool to the types of researchers equipped and excited to utilize such a benchmark.
>
> We hope in this response we have clarified the goals of our original paper and adequately addressed your concerns. Your comments have helped us strengthen the work. If you have any other questions, we would be open to further discussion.
>
> Sincerely,
>
> The authors of Evolution Gym
>
> [1] Du, Tao, et al. "DiffPD: Differentiable Projective Dynamics with Contact." arXiv preprint arXiv:2101.05917 (2021)

---

> ### Author Response · Authors · 2021-08-26
> **Authors' Clarification on the Updated Review**
>
> Dear reviewer j4pt,
>
> Thank you again for your feedback!
>
> We wanted to continue the discussion of the two points you brought up and hear your opinions on how we can improve.
>
> **Number of independent trials**
>
> We agree that more trials lead to more convincing results, however, it’s very common in the robot learning community to use ~6 trials and even fewer. In the following popular benchmark papers, [1-3] use 5 trials, and [4] uses 3-4 trials for evaluation. We believe that 6 guarantees the validity of the result and at the same time is computationally affordable (especially for future researchers who will run this benchmark). This is particularly true given the fact that co-optimization is computationally expensive and we already have many more tasks than existing robot learning benchmarks (for instance, [3] has 10 tasks and [4] has 16 tasks while our benchmark has 30 tasks).
>
> Do you have an alternative suggestion for how many trials you believe would be satisfactory? We would be happy to increase the number of trials to be much greater than 6 in the camera-ready version.
>
> **Scalable complexity and subjective ranking**
>
> We totally agree that being able to design complex environments is important for a benchmark, so we would like to first clarify that complexity in our benchmark is not bounded and can be described quantitatively. Primarily, this can be done by running our tasks on larger robots -- an exponentially larger design space and more actuators create harder co-design problems. In addition, Evolution Gym allows the flexibility for parameterized tasks to be created, for example, passing in as arguments the slope of the terrain, the number of stairs, the locations of objects, etc. Therefore, increasingly complex environments can be easily created by either specifying the larger size of the robot or designing more complex terrains, or targeting harder objectives.
>
> We believe that this ability of our benchmark to scale is one of its advantages. And it is also important to note that it is not common for popular benchmarks in the robot learning community to provide tasks with scalable complexity. For example, in [5,6], the environments are all fixed.
>
> Finally, our current method of ranking tasks is determined by the performance of the co-design baselines (i.e. the number of meaningful solutions the current algorithms can find). But we also want to make clear that evaluation is independent of ranking. The only reason that we provide a ranking is for the reader to get an intuitive sense of the tasks. In addition, given the current state of co-design algorithms, the rankings demonstrate that our benchmark’s tasks are 1) comprehensive 2) of reasonable difficulty. This type of ranking is not uncommon as demonstrated by [3], where tasks were split into two categories by perceived difficulty (SoftGym-Medium and SoftGym-Hard).
>
> Thank you again for your comments and please let us know if you have any remaining concerns.
>
> Sincerely,
>
> The authors of EvoGym
>
> [1] Duan, Yan, et al. “Benchmarking deep reinforcement learning for continuous control.” International conference on machine learning. PMLR, 2016.
>
> [2] Henderson, Peter, et al. “Deep reinforcement learning that matters.” Proceedings of the AAAI conference on artificial intelligence. Vol. 32. No. 1. 2018.
>
> [3] Lin, Xingyu, et al. “SoftGym: Benchmarking Deep Reinforcement Learning for Deformable Object Manipulation.” Conference on Robot Learning. 2020.
>
> [4] Cobbe, Karl, et al. “Leveraging procedural generation to benchmark reinforcement learning.” International conference on machine learning. PMLR, 2020.
>
> [5] Brockman, Greg, et al. “Openai gym.” arXiv preprint arXiv:1606.01540 (2016).
>
> [6] Tassa, Yuval, et al. “Deepmind control suite.” arXiv preprint arXiv:1801.00690 (2018).

---

### Official Review · Reviewer_nQ7Q · 2021-07-16

**Rating:** 8
**Confidence:** 4

**Summary:**

The paper presents Evolution Gym, a benchmark for the evaluation of design and control co-optimization algorithms on simulated voxel-based soft robots. It contains a back-end soft-body simulator (written in C++ with bindings in Python), a set of 30 evaluation tasks of various types and difficulty levels, 3 design optimization algorithms and 1 control optimization algorithm. The algorithms are evaluated on the benchmark tasks and the results are reported in the paper.

**Ethical Concerns:**

No major concerns. See however the section "The discussion on the limitations, perspectives and societal impact of the paper is very short and does not address important points" in the main review.

**Limitations And Societal Impact:**

See the section "The discussion on the limitations, perspectives and societal impact of the paper is very short and does not address important points" in the main review.

**Main Review:**

The paper is very well written and structured. It comes with a well-furnished supplementary including an appendix detailing all the benchmark tasks, a video showing the robot designs and control policies found by the co-optimization algorithms, the source code, as well as a website where the user can visualize all the benchmark tasks.

The diversity of the proposed tasks is impressive and covers a wide range of challenges for design and control co-optimization algorithms. The paper compares the performance of three different design optimization algorithms coupled with a single control optimization algorithm on 10 selected tasks (in the main text, more evaluation in the appendix).

For these reasons, I think the paper is a valuable contribution to the Neurips community. It is important to note, however, that the paper mostly focuses on morphological optimization, control optimization being limited to a single DeepRL algorithm (PPO). As its name indicates, the paper would therefore be more relevant to the Evolutionary Computation community (e.g. in conferences such as GECCO). Nevertheless, it highlights a number of original and interesting challenges that are definitely relevant to Neurips.

My main concerns about the paper are the following (see below for detailled comments on each point):
 - The proposed benchmark imposes a number of constraints that could limit its generality.
 - The discussion on the limitations, perspectives and societal impact of the paper is very short and does not address important points.

I also suggest to the authors to position their contributions wrt to these related papers:
- https://pathak22.github.io/modular-assemblies/resources/assemblies.pdf
- https://arxiv.org/abs/1906.05370
- https://arxiv.org/abs/1810.03779
- https://direct.mit.edu/isal/proceedings/isal2020/32/592/98402

## The proposed benchmark imposes a number of constraints that could limit its generality

Despite its name (Evolution Gym), the proposed benchmark is much more constrained than OpenAI Gym. The main contribution of OpenAI Gym (imo at least) was the proposition of a standardized software interface allowing to easily plug any RL algorithm to any sequential environment. It was completely agnostic to e.g. the definition of observation and action spaces, or to the state space and transition functions of the environments.

Evolution Gym instead imposes a number of hard constraints to the user (at least in its current version):- The environment dynamics is fixed and based on 2D voxel-based soft robots.
- The specification of the robot structure is fixed (Section 3.2).
- Observation and action spaces are mostly pre-specified (section 3.3).
- The co-optimization training procedure is fixed: First the design optimizer proposed a new robot structure to the control optimizer, then the control optimizer computes the optimal controller for the given structure (Section 4).

These constraints could be discussed in more detail in the paper, in particular in the Conclusion:
 - Can the benchmark be easily extended to other types of environments? (i.e. not only voxel-based soft robots, see e.g. the papers mentioned above for possible alternative environments).
 - How can the user define a new task? What amount of code does it involve? What is the software interface? It would be interesting to provide a tutorial in the supplementary material or on the website.
 - Can we evolve/train a robot on a distribution of tasks? (in a meta-learning fashion)
 - Can it be applied to morphogenetic development? (i.e. where the robot structure grows during the lifetime of the robot)
 - Is it possible to easily define other observation and action spaces? What are the limitations in this respect? For instance, is it possible to optimize a decentralized controller? (where each voxel optimizes its own control policy based only on local observation). Or is it possible to specify new sensors, e.g. a light sensor?
- Can the design and the control be jointly optimized? (instead of the currently implemented 2-phase optimization procedure: first design, then control).

It would have been interesting to evaluate more than one control optimization algorithm (currently: PPO). This would have allowed to evaluate interaction effects between design and control optimization (i.e. which control optimization algorithm is better suited for a given design optimization algorithm). Neuroevolution approaches allowing to jointly optimize network architecture and weights (e.g. NEAT or variations of it) would be of particular interest here, as it would challenge the currently imposed 2-phase optimization procedure.

## The discussion on the limitations, perspectives and societal impact of the paper is very short and does not address important points.

- As mentioned above, there are many limitations and perspectives that would deserve to be discussed in the paper.
- There is no mention of societal impact, whereas there are important points to be mentioned here as well (related e.g. on the automatic design of biological creatures)

## Minor comments
- Fig 2: it could help the reader to precise in the caption that the verbal description of the tasks is provided in section 3.5.
- The taxonomy of tasks is not super clear (Locomotion vs. Object Manipulation) for several reasons. First, as mentioned line 191, some tasks are a mix of both. Second, there are other types of tasks in the appendix that are not mentioned in the paper (e.g. shape change). Maybe using labels instead of categories would help (i.e. a task can be associated with multiple labels).
- Lines 327 to 329 suggest that an optimal reward exists for each task. If it is the case, it would be interesting to plot the optimal reward on Fig 3 (and in similar figs in the appendix) in order to better evaluate how well the optimization is able to approach that optimal.
- Lines 298 to 302. This is hard to see this from the Fig, where all but one designs have the mentioned two structures (i.e.  two legs and a structure up top that prevents the block from sliding off).
- Section 5.3: You could say a bit more about how you performed hand design, or at least discuss this point in the discussion or appendix. What procedure did you follow? What are the limitations? What would be a more thorough procedure? (e.g. asking N participants to perform the hand design by imposing some constraints on how to do it, e.g. limited time).
- Regarding the video provided in the supplementary material:
    - I was surprised that most evolved designs are non-symmetrical, even in tasks that are inherently symmetrical like BiderectionalWalker  or Climber. This seems to contradict the claim made in the abstract that robots "often grow to resemble existing natural creatures". It could be interesting to further discuss this point.



**Time Spent Reviewing:**

4

---

> ### Author Response · Authors · 2021-08-10
> **Authors' Reponse to Reviewer nQ7Q**
>
> Dear reviewer,
>
> Thank you for your feedback and constructive suggestions. We are pleased to hear you found our paper well-written and that you see the novelty of our methods. Below, we have addressed your comments in detail.
>
> **(1) Additional References**
>
> Thank you for these great references and we will update the literature review section accordingly.
>
> Ha 2019 [1] focuses on the optimization of continuous parameters of rigid robots. Subsequently, Wang et al. 2019 [2] optimize the morphology of the rigid robots based on a graph representation. Similarly, Pathak et al. 2019 [3] work on graph morphologies of rigid robots with a modularized design space and controller. These works are truly insightful and inspired us when designing our benchmark. Our work can be clearly distinguished from these works because:
> The main contribution of these previous works is algorithmic innovation but our work provides a wide variety of tasks as a comprehensive benchmark, standard baseline implementations, and comparisons of the co-design algorithms.
> Our focus is more on the soft robotics domain which is an emerging area and the research there is relatively under-explored. Our work aims to help accelerate the development of the soft robot co-design field by providing a comprehensive benchmark platform.
>
> Finally, we see Veenstra et al. 2020 [4] as a great complement to our benchmark -- it analyzes the strengths and weaknesses of different morphological encodings. We think it would be really exciting to introduce more types of morphological encoding in Veenstra et al. 2020 [4] to our benchmark and conduct a deeper analysis on our wider variety of tasks.
>
> **(2) Constraints of the Evolution Gym**
>
> We agree that Evolution Gym is certainly not as mature as OpenAI Gym, but Evolution Gym is also designed with flexibility and extendability:
>
> 1) The user is allowed to design new tasks based on our provided simulation. To design a new task, the user can specify the physical specification of the environment, and also customize the definition of the observation space, the action space, and the reward function, which is essentially similar to OpenAI Gym.
> 2) We provide an easy-to-use tool for the user to construct an environment physically (e.g. the ground, floating platforms, soft bridges). With our tool, for example, you can drag the voxel to certain positions and specify its material type easily with our GUI. We will release this tool as well upon acceptance.
> 3) The observation space is not fixed. While we provide standard choices for our benchmark environments, users can definitely customize their own environments with extra observation components.
> 4) The action space depends on the simulation. In our simulation, the robot is actuated by the expansion/contraction of soft actuator voxels. So the action space is fixed given a specific design of the robot, similar to the joint torque/position control in Mujoco.
> 5) For the co-optimization training procedure, it is true that the implemented baseline algorithms share a bi-level optimization routine where the design optimization is in the outer loop while the control optimization is in the inner loop. However, the simulation developed is agnostic to the specific training procedure used and does not constrain the optimization to be bi-level. It actually allows for other types of algorithm procedure on top of it, such as concurrently co-optimizing the design and control, evolving the designs for distribution of tasks, neuroevolution algorithms, or algorithms with decentralized controllers.
>
> Thanks for pointing out these confusing points and we will clarify them in the revised paper. And we also believe that our benchmark can be extended with exciting future works including morphogenetic development, specifying new sensors, evaluating and analyzing different choices of control optimization algorithms.
>
> **(3) Societal Impact**
>
> Thanks for pointing out the fact that we overlooked its potential negative societal impact. We regard this work as a very preliminary piece of research in the field of soft robotics co-design, and we think that we are still far away from causing harm to society. However, we can definitely foresee some problems when this technology can be applied in the real world on a large scale.
>
> As you mentioned, this work may inspire the automatic design of real biological creatures in which serious ethical issues exist. Additionally, since the users have full control over the reward design when customizing the benchmark environments, they could specify pernicious goals and encourage the co-design algorithm to produce more biased results. We will include a more detailed discussion on the negative societal impact in the revised paper.
>
> **(4) Minor Comments**
>
> We are happy to change the various minor comments as suggested as well as clarify our language where appropriate.
>
>
> --------------------------------------------------------------------------------------------------------
>
> Overall, our carefully designed benchmarking tool fills an important missing piece in research in soft robotics and robotic evolution algorithms. We believe NeurIPS would provide the perfect launching pad to expose our tool to the types of researchers equipped and excited to utilize such a benchmark.
>
> We hope in this response we have clarified the goals of our original paper and adequately addressed your concerns. Your comments have helped us strengthen the work. If you have any other questions, we would be open to further discussion.
>
> Sincerely,
>
> The authors of Evolution Gym
>
> [1] Ha, David. "Reinforcement learning for improving agent design." Artificial life 25.4 (2019): 352-365.
>
> [2] Wang, Tingwu, et al. "Neural graph evolution: Towards efficient automatic robot design." arXiv preprint arXiv:1906.05370 (2019).
>
> [3] Pathak, Deepak, et al. "Learning to control self-assembling morphologies: a study of generalization via modularity." arXiv preprint arXiv:1902.05546 (2019).
>
> [4] Veenstra, Frank, and Kyrre Glette. "How Different Encodings Affect Performance and Diversification when Evolving the Morphology and Control of 2D Virtual Creatures." Artificial Life Conference Proceedings. One Rogers Street, Cambridge, MA 02142-1209 USA journals-info@ mit. edu: MIT Press, 2020.

---

> > ### Comment · Reviewer_nQ7Q · 2021-09-01
> > **Thank you for your response**
> >
> > Dear authors,
> >
> > Thank you very much for your response and clarifications.
> >
> > After having read all the other reviews and author replies, here are my updated thoughts about the paper.
> >
> > Most reviewers point out that the benchmark is very much constrained (limited to 2D voxel-based soft robots, with specific observation and action spaces, and a bi-level optimization procedure). The authors agree that there are strong limitations and argue that this doesn't limit the interest of the benchmark because many of the proposed tasks are still not solved by the benchmarked algorithms.
> >
> > I personally think the arguments provided by the authors make sense: the proposed benchmark does evaluate different algorithms on a wide range tasks --some of them unsolved-- and therefore can encourage further contributions in co-optimization algorithms (hence my high score on this paper). However, the authors still need to better convince me that the benchmark is flexible enough to be easily extended (by the authors themselves, but more importantly by other potential contributors). This point is also raised by reviewer j4pt. I don't think it is the case in the current version and the authors haven't convince me yet that it will be the case in the revised version.
> >
> > I suggest to the authors the following concrete improvements:
> >
> > - Describe in detail (e.g. in the supplementary) the interface of each of the main components of the library (abstract classes with their methods and attributes ; and how those classes interact together).
> > - Show that this interface is general enough to easily extend the benchmark in the directions pointed out by the reviewers. This can be done by providing use-case examples on how to replace the simulator by another one (e.g. Mujoco?) ; How to design a new task? ; how to implement new observation and action spaces? ; how to change the training procedure (e.g. co-evolution as part of the same genotype as proposed by reviewer j4pt and myself)?
> >
> > In their reply to my review (and most of the other ones), the authors simply provide general statements such as _"To design a new task, the user can specify the physical specification of the environment, and also customize the definition of the observation space, the action space, and the reward function, which is essentially similar to OpenAI Gym"_ ; or _"While we provide standard choices for our benchmark environments, users can definitely customize their own environments with extra observation components"_. This is not totally convincing to me and I'd like to see more concrete proposals explaining how to extend the benchmark in a practical way. One way to do it would be to provide a set of tutorials (e.g. as Jupyter Notebooks) explaining how to proceed for each of the possible extensions mentioned above.
> >
> > Could you please comment on your opinion about implementing the propositions above in the camera-ready version? I might lower a bit my score if no concrete proposition is made for clarifying the generality and extendability of the library. But I still believe that this paper is a very interesting contribution to the NeurIPS community.
> >
> > Best regards,

---

> > > ### Author Response · Authors · 2021-09-02
> > > **Authors' Reponse to Reviewer nQ7Q**
> > >
> > > Dear reviewer,
> > >
> > > We are glad you see the value of our work, and thank you so much for your ideas. We think they are very helpful and are happy to include them in our camera-ready version. We have detailed them all below, with additional explanations.
> > >
> > > We promise the following concrete changes to the paper:
> > >
> > > **(1) Describe in detail (e.g. in the supplementary) the interface of each of the main components of the library (abstract classes with their methods and attributes; and how those classes interact together).**
> > >
> > > ***Detailed documentation***.  We were already planning on releasing a detailed documentation of our API when we open-source our code so researchers can easily begin experimenting with it. This will include an explanation of classes with their methods and attributes and how those classes interact together. We can also provide starter code with basic experiments and visualization scripts, etc.
> > >
> > > **(2) How to design a new task?**
> > >
> > > ***Tutorials for creating new tasks***. We are planning to add tutorials even though creating tasks is very simple.
> > >
> > > First, we have a GUI for creating simulation objects such as terrain and saving them to file. We can provide a tutorial of how to use this simple interface.
> > >
> > > Additionally, a task is simply a class that extends the Evolution Gym base class and implements an init, step, and reset method. In the init method, terrain objects are loaded into the simulation from a file or array, and the size of the observation/action spaces are specified. In the step method the simulation is stepped. We expose information from the simulation through our API such as positions, velocities, and orientations which are manipulated to compute the observation and reward. The user can access as much or as little exposed information as needed and we can expose more information from the simulation by popular demand as we continue to develop the framework. The reset method resets the simulation using our API and returns an observation. We will provide more detailed explanations in our tutorials and documentation.
> > >
> > > **(3) how to change the training procedure (e.g. co-evolution as part of the same genotype as proposed by reviewer j4pt and myself)**
> > >
> > > ***Tutorials for changing the training procedure***. At the most basic level, our environments can be created, reset, stepped, and closed; just like any other OpenAI Gym environment. The co-evolution strategy you and reviewer j4pt proposed has some non-trivial nuances (like the fact that sizes of the observation and action spaces would vary across robot designs) that would likely make it a nonideal candidate for a tutorial. That being said, if such an algorithm did exist, the create, reset, step, and close functionality would make it easy to integrate with our framework. For the sake of tutorial, we could implement a simple genetic algorithm which could jointly evolve a sinusoidal control and robot design.
> > >
> > > **(4) how to replace the simulator by another one (e.g. Mujoco?)**
> > >
> > > ***Replacing Evolution Gym’s simulator***. Any simulator that supports FEM (finite element method) such as Bullet [1] or any spring-based simulator could be swapped in for our current simulator. Mujoco is a rigid-body simulator so it would not support the soft robots that are used in Evolution Gym. While FEM simulators could be swapped in, they are typically slower than our current simulation method. This is not ideal as we don’t want to spend much time in simulation -- we want to spend more time optimizing the robot/exploring the design space. That being said, we would be happy to provide the option to swap the simulator as we continue to maintain the benchmark, and as co-design algorithms advance.
> > >
> > > ---
> > >
> > > Thank you again for your feedback and let us know if you have any other questions or concerns.
> > >
> > > Sincerely,
> > >
> > > The authors of EvoGym
> > >
> > > [1] Erwin Coumans. 2015. Bullet physics simulation. In ACM SIGGRAPH 2015 Courses. ACM, 7.

---

> > > > ### Comment · Reviewer_nQ7Q · 2021-09-02
> > > > **Reply**
> > > >
> > > > Thank you for all the precisions. I definetly think that adding the extra information you propose will greatly increase the quality of the paper and the acceptance of the benchmark as a standard for the community.
> > > >
> > > > Including a simple genetic algorithm which could jointly evolve a sinusoidal control and robot design is I think a good idea, as it would also allow to compare the results with the current bi-level optimization procedure.
> > > >
> > > > I also recommend to include in the paper the information about which simulator could be compatible or incompatible, which can be important for future users willing to extend the library.
> > > >
> > > > Best,

---

### Official Review · Reviewer_523c · 2021-07-19

**Rating:** 5
**Confidence:** 5

**Summary:**

The paper proposes a benchmark for Evolving Soft Robots. The benchmark is designed to facilitate research in co-design of control algorithms and robot morphologies. The proposed framework relies on a custom physics engine which allows a simple mass-spring-damper connection of individual modules. The proposed benchmark is suitable for both manipulation and locomotion even though restricted to 2D environments only. Authors provide also an initial evaluation of SoTA design optimization algorithms (genetic algorithm, bayesian optmisation, CPPN-NEAT) on the proposed benchmark.

**Ethical Concerns:**

As far as I can tell, no ethical concerns apply on this submission.

**Limitations And Societal Impact:**

I don't see negative potential societal impact for the proposed benchmark.

**Main Review:**

The submitted paper is well written, easy to read and very clear in the writing. To my knowledge, this is the first attempt to provide a benchmark for evaluating co-design on soft robots . Moreover the proposed approach encompasses both locomotion and manipulation. The major limitation of the submitted paper is the significance to the research community. The proposed environment is extremely artificial. First of all it's just a 2D-only environment and a 3D environment would be more appealing for potential adopters of the benchmark. Moreover, the environment doesn't seem suitable to model rotational joints and rotational electric motors which are at the moment the main components for industrial robots.

= MAJOR COMMENTS =

- The proposed environment seem to have important limitations which might hinder adoption from users and therefore the impact on the community. Authors should try to address these limitations or at least conduct experiments to provide evidence that these are not limitations. First limitation is the environment being only 2D; this is a major limitation since applications require 3D modeling. The second limitation is the lack of rotational component;  most of the proposed robot-designs and robot-embodiments are a mixture of soft and rigid components. Third limitation is that authors do not provide evidence that their benchmark can cope with rigid only embodiments; even though soft robotics is an emerging field of research, most of existing industrial applications still rely on assuming rigid only components and testing the physics engine on rigid only robot-designs might attract additional interest. Last but not least, authors have tested the benchmark with a limited number of voxels in the order of 100 voxels; however interesting applications might require orders of magnitude more voxels (1000 to 10000) and no analysis is provided on how the proposed physics engine and computations will scale with the number of voxels.



**Time Spent Reviewing:**

2.5h

---

> ### Author Response · Authors · 2021-08-10
> **Authors' Reponse to Reviewer 523c**
>
> Dear reviewer,
>
> Thank you for your feedback and constructive suggestions. We are pleased to hear you found our paper well-written and that you see the novelty of our methods.
>
> First, we would like to clarify the intended purpose of our work. Our main contribution is to provide the first benchmark environment for robot co-design algorithms. Our benchmark environment is designed to be efficient and effective through selecting a compact robot morphological representation (2D voxel-based soft robot) and implementing a fast simulation (C++-based mass-spring system). Such a lightweight benchmark helps reduce the time spent in the simulation during evaluating AI algorithms and accelerates the development iteration of the new algorithms. We hope that future researchers will use Evolution Gym as a tool to develop new, powerful co-design techniques that could not have been realized otherwise.
>
> Below, we have addressed your major comments in detail.
>
> **(1.1) The environment is 2D. Most applications require 3D modeling.**
>
> Thanks for pointing out this. Our benchmark is designed to be a quick testbed for co-design algorithms. However, the co-design algorithms can be agnostic to whether the application is 2D or 3D (similarly as reinforcement learning algorithms for control optimization which are agnostic to whether the problem to be solved is 2D or 3D).
>
> Further, we believe the 2D nature of our environment is one of its strengths. 2D soft-body simulation provides enough complexity of the task while being much more computationally cheap compared to 3D soft-body counterparts. For instance, training a controller for a 3D soft robot on a simple task can take on the order of hours based on more realistic FEM simulation (see Du et al. 2021 [1]). By comparison, in our framework based on 2D mass-spring systems, the control policy of a single robot can be trained on a 4-core machine in a matter of minutes. With such speeds, we believe that future research endeavors in co-design and control optimization algorithms would be more computationally tractable.
>
> Finally, as evidenced by our experimental results, the state-of-the-art co-optimization algorithms fail to search successful robots on many of our hard tasks, even in a simple 2D environment. In other words, the limitations of the learning algorithms stopped us from increasing the sophistication of our environment. However, we believe that 3D extension is an important future work when the learning algorithms become more intelligent and sample efficient.
>
> **(1.2) Rigid only embodiments and no rotational component.**
>
> We focus on voxel-based soft robots in this work with inspiration from real cell-based organisms in nature. Furthermore, soft robotics is an emerging area of research and it provides enough complexity and challenges to evaluate various co-design algorithms. The goal of this work is to provide a fast platform to develop and analyze co-design algorithms that can later be used in a more sophisticated and time-consuming system for real applications. To be noted, although rigid and rotational components are not explicitly incorporated in our simulation, they can be implicitly represented by our system. More specifically, the rigid component can be realized by using cells with stiff spring materials which is also a common practice in soft robotic simulation to approximate rigid components. Rotational actuator structures can also be formed by a mixture of different soft cells. As evidence, some interesting rotational structures are evolved to be functional to rotate the legs of the robots in our walking task. We agree that explicitly supporting rigid and rotational components is an interesting and practical addition to our system and would generate more interesting robot designs, and would be exciting future work.
>
> **(2) Scalability of the physics engine as the number of voxels increases.**
>
> We describe a quick experiment that demonstrates our simulation can scale to this size; we will provide a more rigorous analysis in our revised paper. The following describes the result of running an (N x N) robot actuating randomly in our simulation. The simulation was run on a single core of a standard laptop with an i7 processor. We report average simulation steps per second (ASSPS).
>
> - N = 5, #voxels = 25, ASSPS = 239, #voxels x ASSPS = 5975
> - N = 10, #voxels = 100, ASSPS = 146, #voxels x ASSPS = 14600
> - N = 20, #voxels = 400, ASSPS = 62, #voxels x ASSPS = 24800
> - N = 30, #voxels = 900, ASSPS = 31, #voxels x ASSPS = 27900
>
> We totally agree that robots with larger numbers of voxels will be an important feature and the results above show that our simulation is capable of simulating and evaluating large robots. In our current work, we limited the scope of the problem to small-sized designs since we found that it already poses big challenges to the current state-of-the-art algorithms. As shown in the results section, current state-of-the-art algorithms failed to discover feasible 25-50 voxel solutions on some of our tasks. We believe that our proposed benchmark can help advance this field and novel algorithms can be invented to solve for larger robot design space in the future.
>
> ----------------------------------------------------------------------------------------------------------
>
> Overall, our carefully designed benchmarking tool fills an important missing piece in research in soft robotics and robotic evolution algorithms. Armed with the flexible and expressive framework EvoGym provides, we are optimistic that future researchers will use Evolution Gym to improve co-design methods, not just by tweaking existing algorithms slightly, but by fundamentally redesigning them. We believe NeurIPS would provide the perfect launching pad to expose our tool to the types of researchers equipped and excited to utilize such a benchmark.
>
> We hope in this response we have clarified the goals of our original paper and adequately addressed your concerns. Your comments have helped us strengthen the work. If you have any other questions, we would be open to further discussion.
>
> Sincerely,
>
> The authors of Evolution Gym
>
> [1] Du, Tao, et al. "DiffPD: Differentiable Projective Dynamics with Contact." arXiv preprint arXiv:2101.05917 (2021)

---

### Decision · Program_Chairs · 2021-09-28

**Decision:**

Accept (Poster)

**Comment:**

Meta-review for "Evolution Gym: A Large-Scale Benchmark for Evolving Soft Robots"

While policy optimization is a well-studied topic in both control and RL, little work has been done on robot-design optimization, in particular, joint optimization of the robot agent and its policy. This work is motivated at filling the gap, and proposes a benchmark for evolving (both policy and controller) of soft-bodied robots (a rich area in evolutionary computation, but rather less explored in the RL community). Rather than rely on detailed and computationally expensive simulation, they opted for simplicity and created a benchmark using a simple, but fast 2D physics engine (a mass-spring system in C++) to implement a suite of soft-bodied robot environments with OpenAI-Gym like interface. They provide sufficient details on the assumptions and simplifications made in the simulation. See Supplementary Materials for examples: https://sites.google.com/view/evolution-gym-benchmark/all-tasks/object-manipulation

Reviews were mixed, with two reviewers giving 4’s and 5’s, and two reviewers giving 7 and 8’s. That being said, all reviews agreed that the paper is well written, and is a novel attempt at providing a benchmark for evaluating soft-robots.

The main criticism of the work raised is that the current environment is not realistic enough to be useful for real robot design, hence the significance to certain parts of the research community is reduced. The counter argument raised by positive reviewers (and authors) is that the simplification is actually intentional, and that everyone including authors agree that the proposed benchmark is very much constrained (limited to 2D voxel-based soft robots, with specific observation and action spaces). As one reviewer put it: *The authors agree that there are strong limitations and argue that this doesn't limit the interest of the benchmark because many of the proposed tasks are still not solved by the benchmarked algorithms.*

To improve the work, reviewers have proposed that the authors address the simplicity by not necessarily making it more realistic and useful for realistic robot design, but rather keep the simplicity, but make the framework really extensible, so it can easily be extended as needed by users. For instance as raised by nQ7Q, *One way to do it would be to provide a set of tutorials (e.g. as Jupyter Notebooks) explaining how to proceed for each of the possible extensions mentioned above. Show that this interface is general enough to easily extend the benchmark in the directions pointed out by the reviewers. This can be done by providing use-case examples on how to replace the simulator by another one (e.g. Mujoco?) ; How to design a new task? ; how to implement new observation and action spaces? ; how to change the training procedure?* The authors have acknowledged the suggestion and opted to provide comprehensive tutorials and detailed API documentation.

At the end of the day, discussions from both sides didn't converge, and while reviewers agreed to disagree, they recognized each other's points. While the current benchmark is likely not going to be of any direct use for designing real world physical robots, the proposed benchmark is a solid attempt at providing the first co-design benchmark, a break from the traditional pure policy optimization RL benchmarks, and despite its simplicity, the tasks proposed still cannot be solved by current methods. Arguably, like the original OpenAI gym, the simplicity, and speed of execution might outweigh direct real world applications, and instead help grow the co-design research community.

In this AC's view, there are already numerous Gym environments that also emphasize on photorealism and high quality realistic physics simulations (including accurate fluid and mesh simulations), but those environments haven't really gained traction from the community possibly due to the high barrier to entry in terms of compute, slowing down research iteration, leading us to be where we are in RL. The RL community (perhaps not the robotics community) will have something to gain from this co-design benchmark, which may help facilitate a move towards solving problems that is not pure policy-optimization, and perhaps this is what's needed to move RL research into new directions to help break away from the local optimum we are hill-climbing together as a community. Furthermore, the benchmark may act as a bridge for the NeurIPS RL community and the evolution / co-design folks and help facilitate a common language / platform to exchange ideas, results and discussion. In my view, the speed and simplicity will also help improve accessibility to folks without access to SOTA compute facilities, which may help bring in additional talent in (as we have seen with OpenAI Gym).

For these reasons, I'm recommending acceptance of this work, despite the borderline score of 6, because I do believe the benchmark will be beneficial to the NeurIPS community at large.


**Consistency Experiment:**

NeurIPS has a long history of experimentation. In 2014, NeurIPS ran an experiment in which 10% of submissions were reviewed by two independent committees to quantify the randomness in the review process. This year, we repeated a variant of this experiment to see how the quality of the review process has changed over time.  This paper was part of the experiment and was therefore assigned to two committees (consisting of reviewers, an Area Chair, and a Senior Area Chair) that reached independent decisions.  If both committees made the same recommendation, this recommendation was followed. If a single committee recommended acceptance, the paper was accepted (with the exception of a few cases in which the other committee identified what we considered a fatal flaw, e.g., an error in a key result).

This copy’s committee reached the following decision: **Accept (Poster)**

The other committee assigned to the paper recommended **Reject**.  You can find the other set of reviews, along with any follow up discussion with the authors here:
https://openreview.net/forum?id=lM2971LAwV